# Differences between *Mycobacterium chimaera* and *tuberculosis* Using Ocular Multimodal Imaging: A Systematic Review

**DOI:** 10.3390/jcm10214880

**Published:** 2021-10-23

**Authors:** Sandrine Anne Zweifel, Nastasia Foa, Maximilian Robert Justus Wiest, Adriano Carnevali, Katarzyna Zaluska-Ogryzek, Robert Rejdak, Mario Damiano Toro

**Affiliations:** 1Department of Ophthalmology, University Hospital Zurich, Frauenklinikstrasse 24, 8091 Zurich, Switzerland; foa.nastasia@gmail.com (N.F.); maximilian.wiest@usz.ch (M.R.J.W.); toro.mario@email.it (M.D.T.); 2University of Zurich, 8091 Zurich, Switzerland; 3Department of Ophthalmology, University Magna Grecia of Catanzaro, 88100 Catanzaro, Italy; adrianocarnevali@unicz.it; 4Department of Pathophysiology, Medical University of Lublin, 20090 Lublin, Poland; katarzyna.zaluska-ogryzek@umlub.pl; 5Chair and Department of General and Pediatric Ophthalmology, Medical University of Lublin, 20079 Lublin, Poland; robertrejdak@yahoo.com

**Keywords:** mycobacterium, multifocal choroiditis, fundus photography, biomicroscopy, multimodal imaging, fundus fluorescein angiography, indocyanine green angiography, fundus autofluorescence, optical coherence tomography angiography, optical coherence tomography

## Abstract

Due to their non-specific diagnostic patterns of ocular infection, differential diagnosis between *Mycobacterium* (*M.*) *chimaera* and *tuberculosis* can be challenging. In both disorders, ocular manifestation can be the first sign of a systemic infection, and a delayed diagnosis might reduce the response to treatment leading to negative outcomes. Thus, it becomes imperative to distinguish chorioretinal lesions associated with *M. chimaera*, from lesions due to *M. tuberculosis* and other infectious disorders. To date, multimodal non-invasive imaging modalities that include ultra-wide field fundus photography, fluorescein and indocyanine green angiography, optical coherence tomography and optical coherence tomography angiography, facilitate in vivo examination of retinal and choroidal tissues, enabling early diagnosis, monitoring treatment response, and relapse detection. This approach is crucial to differentiate between active and inactive ocular disease, and guides clinicians in their decisional-tree during the patients’ follow-up. In this review, we summarized and compared the available literature on multimodal imaging data of *M. chimaera* infection and tuberculosis, emphasizing similarities and differences in imaging patterns between these two entities and highlighting the relevance of multimodal imaging in the management of the infections.

## 1. Introduction

Mycobacterium (M.) belongs to the genus of Actinobacteria, which consists of over 190 species, some of them causing serious health diseases in humans [1]. The most common pathogen is *M. tuberculosis*, which has a panoply of systemic manifestations, including ocular manifestations [1,2]. First discovered in 1882 by Robert Koch, *M. tuberculosis* is an intracellular slow-growing pathogen transmitted via the inhalation of aerosolized, bacteria-containing droplets. It is the etiologic agent of tuberculosis (TB), which remains a significant global public health burden [2,3,4].

In the last few years, *Mycobacterium chimaera*, a non-tuberculous mycobacterium (NTM), has gained awareness because of a worldwide hospital-acquired outbreak of disseminated infections following open-heart surgery [5,6,7,8,9,10]. *M. chimaera* is a non-tuberculous, slow-growing, mycobacterium that belongs to the Mycobacterium avium complex (MAC) [6,11,12]. Before its first identification, owing to ambiguous genetic features characterizing its strains, it had been wrongly identified as *M. avium*, *M. scrofulaceum*, or *M. intracellulare* by most of the clinical microbiology laboratories [5,11]. *M. chimaera* acts as an opportunistic pathogen. Before 2013, this mycobacterium was known to cause respiratory infections in immunocompromised individuals [5,6]. Since 2013, there has been an unprecedented increase in the incidence of *M. chimaera* infections, especially among individuals undergoing cardiopulmonary bypass surgery. It is primarily transmitted via thermoregulatory components of extracorporeal membrane oxygenation (ECMO) systems and water tanks of heater-cooler units (HCUs) [5,6]. With increased usage of ECMO systems and the ability to distinguish *M. chimaera* from other MAC bacteria, more attention has been given to this strain [5,11]. The first cases were observed in Switzerland in 2013. However, since then, *M. chimaera* affected cases have been reported in 11 other countries, including Canada, the USA, Europe, and Australia [11,12].

Both *M. chimaera* and *M. tuberculosis* cause disseminated infections, inducing granulomatous inflammation in multiple organs [6,13]. Tuberculosis (TB) generally affects the lungs, but can also affect other parts of the body. Only 10% of TB infections are symptomatic, manifested as a chronic cough along with fever and weight loss. If other organs are involved, a wide range of extra-pulmonary manifestations can also occur [4,13]. For example, ocular TB is a rare extrapulmonary manifestation of TB infection and a great imitator of other ocular pathologies [2]. The reported prevalence of tubercular uveitis varies widely, ranging from 0.2–2.7% in non-endemic regions to 5.6–10.5% in highly endemic regions [6,14,15]. If left untreated, progressive tuberculous infections kill about 50% of those affected [4]. Systemic symptoms and laboratory features of *M. chimaera* infections are also often non-specific and include unexplained low-grade fever, fatigue and sometimes also dyspnea [6]. Similarly, to a TB infection, if other organs are infected, additional organ-specific symptoms can occur [5,13,16,17]. In particular, it has been shown that the infection has a very strong proclivity for the eyes [7,8,9,10,18]. If not promptly diagnosed and treated, *M. chimaera* infections often become life-threatening [5,7,9].

While for TB infections, ocular and extraocular involvements have been described (including uveitis, retinal-choroidal lesions, optic neuropathy and endophthalmitis but also orbital, corneal, scleral, eyelid, conjunctival and lacrimal gland involvements) [2], for *M. chimaera* the most common ocular manifestation are chorioretinal lesions; mild anterior and intermediate uveitis and optic disc swelling. Secondary complications include macular and retinal neovascularization. Zweifel et al. [7,9] have demonstrated that choroidal manifestation lesions seem to reflect systemic disease activity and can be used as an early diagnostic biomarker for assessment of treatment efficacy [6,7,8,9,10].

To date, limited data exists about *M. chimaera* infections, a rare and often-lethal disease. Ophthalmologists should be aware of this recently described entity and its systemic and ocular findings. They may play a key role for differential diagnosis, for monitoring treatment response and detection of recurrences after discontinuation or adjustment of treatment due to adverse drug effect.

Differential diagnosis of ocular lesions can be often challenging, thus in this review we summarized and compared the available literature on ocular multimodal imaging data of *M. chimaera* and TB, emphasizing similarities and differences in imaging patterns between these two entities and highlighting the relevance of multimodal imaging in the management of the infections.

## 2. Materials and Methods

### 2.1. Search Methods for Identification of Studies

This systematic review was conducted and reported in accordance with the preferred reporting items for systematic reviews and meta-analyses guidelines [14]. The review protocol was not recorded at study design, and no registration number is available for consultation.

The methodology used for this comprehensive review consisted of a systematic search of all available articles exploring the current available diagnostic tools on *M. tuberculosis* and *chimaera*.

A comprehensive literature search of all original articles published up to August 2021 was performed in parallel by two authors (S.Z. and M.D.T.) using the PubMed, Cochrane library, Embase, and Scopus databases.

For the search strategy, we used the following keywords and Mesh terms: “*M. tuberculosis*”, “*M. chimaera*”, “Fundus photography”, “Biomicroscopy”, Fundus Fluorescein angiography”, Indocyanine green angiography”, “Fundus autofluorescence”, “Ocular coherence tomography”, and “Optical coherence tomography angiography”. Furthermore, the reference lists of all identified articles were examined manually to identify any potential studies that were not captured by the electronic searches.

The search workflow was designed in adherence to the preferred reporting items for systematic reviews and meta-analyses (PRISMA) statement and according previous reports [14,15,16,17,19,20].

### 2.2. Eligibility Criteria

All studies available in the literature, reporting on original data on *M. tuberculosis* and *chimaera* infection with ocular involvement, were included, without restriction for study design, sample size, and intervention performed. Review articles or articles not written in English were excluded.

### 2.3. Data Collection

After preparation of the list of all electronic data captured, two reviewers (N.F. and M.R.J.W.) examined the titles and abstracts independently and identified relevant articles according to the eligibility criteria. Any disagreement was assessed by consensus and a third reviewer (A.C.) was consulted when necessary.

The reference lists of the analyzed articles were also considered as potential sources of information. For unpublished data, no effort was made to contact the corresponding authors.

## 3. Results

The results of the search strategy are summarized in Figure 1. From 510 articles extracted from the initial research, 425 abstracts were identified for screening and 41 of these met the inclusion/exclusion criteria for full-text review (Figure 1).

No data synthesis was possible for the heterogeneity of available data and the design of the available studies (i.e., case reports or case series). Thus, the current systematic review reports a qualitative analysis, detailed issue-by-issue below in narrative fashion.

### 3.1. Fundus Photography and Biomicroscopy

Using both fundus photography and biomicroscopy, chorioretinal lesions can be detected among patients infected by *M. chimaera*, but the extent of these lesions can vary widely [7,9,10]. Some patients have few inactive choroidal lesions, but others develop widespread lesions (progressive ocular disease). The lesions appear white/yellowish and usually have a round or ovoid shape. At the beginning they are usually small (50–300 mm diameter), but their size typically increases over 4 to 8 weeks in active disease. The lesions appear diffusely over both the posterior pole and the retinal periphery and have a particular uniform distribution [7,9,10]. In few patients, a stable period with clinically inactive lesions before progression to active ocular disease can occur. Indistinct borders on fundus photography represent a sign of activity, whereas inactive or quiescent patients have well-defined borders and appear as “punched-out” lesions. Active and inactive lesions, as well as atrophic lesions, can be observed close to each other in the same eye in patients with active ocular disease. Inactive lesions should be monitored as well, as many of them remain quiescent over time, however, after adaptation/stopping of antimycobacterial treatment, recurrence can be observed [9]. Nevertheless, the activity status of *M. chimaera* infection should not be determined only on the basis of biomicroscopy or color fundus photography findings. A comprehensive ophthalmological examination using multimodal imaging is required. Indeed, the diagnosis and monitoring of *M. chimaera* choroiditis require a detailed ophthalmic including wide-angle fundus photography, FA/ICG (if possible, using a wide-angle camera), EDI OCT, FAF and OCTA (if available) as proposed by Böni et al., should be performed [9].

Biomicroscopy can be used to examine the anterior segment masses, single choroidal masses, or multifocal lesions, serpiginous-like choroiditis (SLC) and retinal vasculitis in intraocular TB patients [21,22]. Tubercle granulomas appear as ill-defined, yellowish, round-to-oval nodules that primarily involve the posterior pol. Furthermore, multiple, non-contiguous choroiditis might develop into a contiguous, diffused pattern called a serpiginous-like lesion, since it looks like serpiginous choroiditis. However, unlike serpiginous choroiditis, the serpiginous-like lesions tend to spare the fovea, are multifocal, and non-contiguous to the optic disc. In addition, unlike serpiginous choroiditis, the tuberculous serpiginous-like choroiditis (TB-SLC) cases exhibit an inflamed vitreous [2,23]. Usually, SLC develops in early to middle aged individuals. Choroidal granulomas are usually unilateral, large, solitary, elevated yellowish subretinal masses resembling a tumor. These granulomas are usually associated with an overlying exudative retinal detachment and located in the posterior fundus.

However, the diagnosis of ocular TB is still challenging, owing to the involvement of mixed tissues and heterogenous infections [24]. The emergence of more advanced TB diagnostic tools, such as IFN-γ release assays, radiodiagnostics, and molecular biology techniques has markedly improved the specificity of ocular TB diagnosis [25]. However, the sensitivity and specificity of current diagnostic tools is still suboptimal, which delays ocular TB diagnosis and treatment. The diagnosis of ocular TB is confirmed only when the causal bacteria are isolated from the ocular tissue. The status of presumed TB uveitis is determined if any of the following signs is observed: choroidal granuloma, retinal vasculitis with or without choroiditis, broad-based posterior synechiae, or serpiginous-like choroiditis (SLC) with a positive tuberculin skin test or QuantiFERON-TB Gold test, or any other relevant tests [26].

The ocular *M. chimaera* infection-induced chorioretinal lesions resemble ocular TB-induced multifocal choroiditis [27,28]. However, *M. chimaera* infection-induced lesions are uniformly distributed and rounded, whereas the TB-induced lesions are irregularly distributed and have variable shapes [9]. In addition, although ocular TB is manifested in several ways, none of the ocular *M. chimaera* cases exhibit any of these manifestations that are indicative of TB.

### 3.2. Fundus Fluorescein Angiography (FA)

FA shows potential in differentiating active from inactive disease, characterizing choroidal, optic disc, and retinal involvement, detecting complications, and monitoring treatment response [9,23].

The FA analysis of *M. chimaera*-infected cases shows the presence of active choroidal lesions that are hypofluorescent in early frames and stained in the later ones. However, early phase of FA usually shows hyperfluorescence of inactive lesions with hyperfluorescence in late phase according to a window defect. The fluorescent pattern for active and inactive lesions seems to be similar for both diseases. However, the shape and type of lesions, and also their distribution, differs between the two entities [9,22,24]. Contrary to the TB cases, the *M. chimaera*-infected cases do not present capillary nonperfusion, cystoid macular edema, or retinal vasculitis [7,9,10].

FA of choroidal granulomas and tubercles in ocular TB reveals active tubercles with hypofluorescence in early phases and hyperfluorescence in late phases, and inactive tubercles (healed) that exhibit transmission hyperfluorescence [23]. Large choroidal granulomas might exhibit hyperfluorescence in early phases and a dilated capillary bed, followed by a progressive increase in hyperfluorescence, and finally, pooling of dye in the subretinal space in the late phase. FA can also be used to diagnose retinal angiomatous proliferation (type 3 macular neovascularization) that develops alongside acute choroidal tubercle or granuloma and might require intravitreal antivascular endothelial growth factor (VEGF) therapy along with the usual therapy [23]. In SLC patients, FA reveals hypofluorescence in early phases and hyperfluorescence in the late phases. The lesions that are healed might exhibit either transmission hyperfluorescence or hypofluorescence in the early phase and staining in the late phase owing to late scleral staining through the thin choroid. During FA analysis, active lesions exhibit hypofluorescence in early phases with hyperfluorescence in late stages, while the inactive lesions exhibit hypofluorescence towards the center and hyperfluorescence towards the periphery [23]. Tuberculous retinal vasculitis is another manifestation of ocular TB. It primarily affects the veins and is occlusive in nature. Active vasculitis manifests in the form of severe perivenular cuffing with thick exudates, and is generally associated with focal choroiditis lesions, moderate vitritis, and retinal hemorrhages [16,24,27]. Occlusive retinal vasculitis might even lead to retinal neovascularization. Furthermore, this technique can also be used to diagnose cystoid macular edema, which is characterized by dye leakage and accumulation in cystic spaces around fovea, in a “petaloid” pattern, or to diagnose macular neovascularization that emerges at the border of an inactive choroidal lesion or optic disc neovascularization [22,24,25,27,28].

Clinically, tuberculous optic neuropathy is manifested as papillitis, retrobulbar optic neuropathy, and neuroretinitis. In cases with neuroretinitis or papillitis, FA reveals optic disc hyperfluorescence in early phases followed by leakage during the late phases [23].

The FA presentation of the chorioretinal lesions due to ocular *M. chimaera* is similar to multifocal choroiditis related to ocular TB. The fluorescent pattern for active and inactive lesions seem to be similar for both diseases. However, the shape and the type of lesions, and their distribution, differs between the two entities [9,22,24]. Additionally, retinal vasculitis was not observed in any of the patients with disseminated *M. chimaera infection*.

These aspects are summarized in Table 1, Figure 2 and Figure 3.

### 3.3. Indocyanine Green Angiography (ICGA)

Because of its unique properties, ICGA lends itself to study the choroidal vasculature. In *M. chimaera*-infected eyes, multiple hypocyanescent lesions can be detected in ICGA in both early and late phases, many more compared to fundus photography or FA. ICG reveals the total lesion number and is, therefore, better suited to detect overall disease progression.

During the late phases of the ICGA examination, compared to inactive lesions, active lesions exhibit a stronger hypocyanescence for a longer duration. However, hypocyanescence of the lesions usually does not regress completely, even in clinically quiescent patients with clearly demarcated lesions on fundus photography. The persistence of hypocyanescence of these lesions is probably due to atrophy of the choriocapillaris/stromal choroid [9].

In TB posterior uveitis cases, ICGA reveals the presence of tubercular choroidal granulomas in choroidal stroma in the form of rounded hypocyanescent lesions during the dye transit and isocyanent lesions in the later phase [29,30]. In cases where the granuloma is present across the whole thickness of choroidal stroma, the lesions appear hypocyanescent throughout the course of the analysis. More than 90% of tuberculous granulomas are full thickness. Those granulomas have the same hypocyanescent behavior as *M. chimaera* lesions. As in multifocal choroiditis due to *M. chimaera*, most of the time, the tubercular granulomas visible on ICGA correspond to the hyporeflective lesions visible on EDI-OCT [23,31].

Compared to FA, ICGA is more efficient in examining the TB-SLC lesions. Even very subtle lesions can be identified on ICGA [22,29,30,32]. This is due to the involvement of the retinal pigment epithelium cells (RPE) that occurs later during the course of the disease. The SLC lesions are hypothesized to develop due to the occlusion of choriocapillaris that are visible in the form of irregularly-shaped hypocyanescent lesions during the early phase of the dye transit. They are still visible as hypocyanescent lesions during the late phases, because choriocapillaris are occluded and retain the dye. The healed stage is characterized on ICGA by better delineation of atrophic choroid [28,32].

### 3.4. Fundus Autofluorescence (FAF)

When using FAF in *M. chimaera*-infected eyes, active chorioretinal lesions appear as hyperautofluorescent areas in patients with progressive disease, whereas inactive lesions are usually hypoautofluorescent [7,9]. The hypoautofluorescence of the lesions is indicative of death of RPE and overlying photoreceptors. On the other hand, hyperautofluorescence of the lesions is indicative of loss of function. As in fundus photography, new active hyperautofluorescent lesions can often be observed close to atrophic hypoautofluorescent lesions. In the very late stages, lesions frequently show a confluent hypoautofluorescence, indicating a convergent loss of RPE [9].

Few studies have focused on the FAF patterns of TB granulomas. Gupta et al. [33] classified TB-SLC progression into four stages.

Stage I: Acute lesions; poorly-demarcated amorphous hyperautofluorescence halo; lasts for 2 to 4 weeksStage II: Lesions with a well-demarcated hypoautofluorescent border; lasts for 6 to 8 weeksStage III: Lesions with mottled hypoautofluorescence and dispersed granular hyperautofluorescenceStage IV: Completely healed lesions with uniform hypoautofluorescence

The initial hyperautofluorescence has been attributed to the choroidal inflammation-induced fluorophore accumulation in the retina [34]. FAF has emerged as a powerful tool to monitor the TB serpiginoid lesion progression. It is advantageous over conventional angiography since the conventional technique is invasive and makes it difficult to interpret the results when the lesions comprise both active and inactive elements. FAF is also more efficient than ICGA, since ICGA cannot demarcate between healed and active lesions. Moreover, FAF can also provide the status of the RPE-photoreceptor complex [23].

### 3.5. Ocular Coherence Tomography (OCT)

In cases with progressive *M. chimaera* infection, enhanced depth imaging ocular coherence tomography (EDI-OCT) reveals rounded hyporeflective areas with significantly decreased choroidal vascularity. In EDI-OCT, the active lesions are generally well-demarcated and well correlated with hyporeflective regions, whereas inactive lesions are poorly demarcated and do not always correlate with hyporeflective regions. The thick lesions impact the adjoining RPE, making it irregularly shaped and attenuated. Choroidal hypertransmission can be observed in both inactive and active lesion. Choroidal lesions often underlie ellipsoid zones with altered integrity. In a previous case study, progressively large granuloma led to a Type 2 macular neovascularization with subretinal fluid development and concomitant visual decrease [9].

To date, no studies regarding the anterior segment OCT in patients with *M. chimaera* infection are available. In ocular TB, OCT of the anterior segment reveals a poorly-demarcated amorphous lesion, and synechiae and narrowing of anterior chamber exudates, corneal edema, and iridocorneal angle. The treatment of the condition usually leads to corneal edema regression and corneal thickness and exudate reduction [35].

In the case of intraretinal TB granuloma, OCT reveals a rounded, hyperreflective lesion in the neurosensory retina, which comprises a partially hyporeflective core underlying a hyperreflective area with surrounding neurosensory retinal detachment [36]. The presence of hyperreflective dots in the outer retina are attributed to proliferating RPE. Non-homogenous localized thickening is observed in the RPE/choriocapillaris complex under the retinal lesion devoid of any dome-shaped retinal elevation [23,36].

While describing the OCT characteristics of tuberculous choroidal granulomas, Salman et al. reported a local adhesion area (known as “contact sign”) between the RPE–choriocapillaris layer and the overlying neurosensory retina [37]. Another study revealed the presence of nonhomogeneous lobulated patterns in tubercular choroidal granulomas [38]. This finding could be instrumental in the identification of smaller granulomas and in the differentiation between granulomas and normal large choroidal vessel lumens [39]. The proliferating RPE cells and granularity of outer photoreceptor layer reportedly contribute to the pathophysiology of choroidal granulomas [40,41].

In a previous study, EDI-OCT revealed choroidal infiltration with RPE elevation in lesion regions of TB-SLC patients [42]. In addition, SD-OCT can be used to reveal differences between active multifocal choroiditis and serpiginous choroiditis. In cases with serpiginous choroiditis, the structural alterations are limited to the outer retina. On the other hand, active multifocal choroiditis lesion is characterized by an increase in the reflectivity of inner retina [42]. Post-choroiditis treatment, the healed lesions exhibited increased reflectivity of only the outer retina region.

In the active choroiditis phase, there is either no change or a slight increase in the thickness of the retina; however, its thickness decreases mildly after the lesion has healed, which might be attributed to retinal atrophy [41]. With an increase in the thickness of the choroid, its reflectivity increases (also known as “waterfall effect”), which is attributed to inflammatory cell infiltration [43]. In a previous study, Bansal et al. [44] described the SD-OCT changes in the outer retina of an SLC patient. Their findings showed a correlation between the OCT and FAF findings. For instance, the spread of hyperreflectivity into the outer retina on OCT corresponded to the alterations in the photoreceptor outer segment tips, the RPE layers, and photoreceptor ellipsoid and myoid junction. They also reported a thickening of RPE/Bruch’s membrane complex in the areas with lesions [44].

Furthermore, OCT can be used for the diagnosis and monitoring of uveitic macular edema too. Individuals with presumed ocular TB exhibit varied patterns of macular edema, such as serous retinal detachment, diffuse macular edema, and cystoid macular edema. The central macular thicknesses observed on OCT can be used as an indicator of treatment response [45].

### 3.6. Optical Coherence Tomography Angiography (OCTA)

Before the emergence of OCTA, ICGA was the only imaging modality available for the assessment of choroidal circulation. OCTA proved to be highly efficient non-invasive imaging technique that could be used to attain high resolution in vivo images. With disease progression, a reduction in the flow could be observed in the choriocapillaris/inner choroid image obtained using OCTA for the regions containing lesions observed in the images from ICGA. Similar to ICGA, OCTA could detect a higher number of lesions compared to that detected using other conventional techniques, such as FA and fundus photography. Böni et al. [9] reported that OCTA was able to reveal neovascular flow overlying a large chorioretinal lesion in their case study. The OCTA image corroborated the presence of neovascular flow in the area of subretinal hyperreflective material (SHRM), as detected using the structural OCT. The OCTA image of the neovascular lesion revealed the presence of small- and medium-caliber vessels with branched tiny vessels and arcades at vessel termini. Intravitreal anti–vascular endothelial growth factor (anti-VEGF) treatment was administered, which led to significant amelioration of the neovascular membrane. Follow-up assessment revealed a reduction in retinal thickness, SHRM, and area and density of the neovascular membrane [9].

Intraocular TB induces an inflammatory choriocapillopathy [37,40]. As in *M. chimaera* eyes, OCTA reveals a choriocapillaris flow reduction in the corresponding area of the ICGA-detected lesions in TB-SLC patients. These regions appear as well-delineated hyporeflective regions dispersed with few choriocapillaris spots towards the center of the lesion. The affected regions could be more precisely determined using OCTA than ICGA. During lesion healing, some patients may develop choriocapillaris atrophy [46,47].

In addition, OCTA facilitates vascular abnormality detection among these patients [48]. Previously, OCTA was used for the detection of macular neovascularization in a TB-SLC patient [49]. In this study, OCTA could efficiently differentiate between the neovascular membrane and the various retinochoroidal layers of the abnormal vascular network [49]. OCTA has also been deemed to be useful to monitor TB-SLC progression and its worsening post-therapy (unpublished data). Taken together, OCTA holds great potential in the diagnosis and management of posterior uveitic entities. Future studies with large sample sizes using OCTA could further elucidate the pathophysiology of this disease [23,49].

## 4. Conclusions

*M. chimaera* and *M. tuberculosis* infections can be very aggressive and can lead to death. A timely diagnosis of those infections is not always easy. Multimodal imaging of ocular findings can lead to an early diagnosis and a prompt initiation of anti-tuberculosis therapy, preventing poor patient outcomes. Furthermore, it gives the possibility to differentiate active and inactive ocular lesion and, because of the likely association with systemic disease activity, it plays a critical role for monitoring patients under treatment and for evaluation in the follow-up after discontinuation of treatment or treatment adjustment due to adverse drug effect.

## Figures and Tables

**Figure 1 jcm-10-04880-f001:**
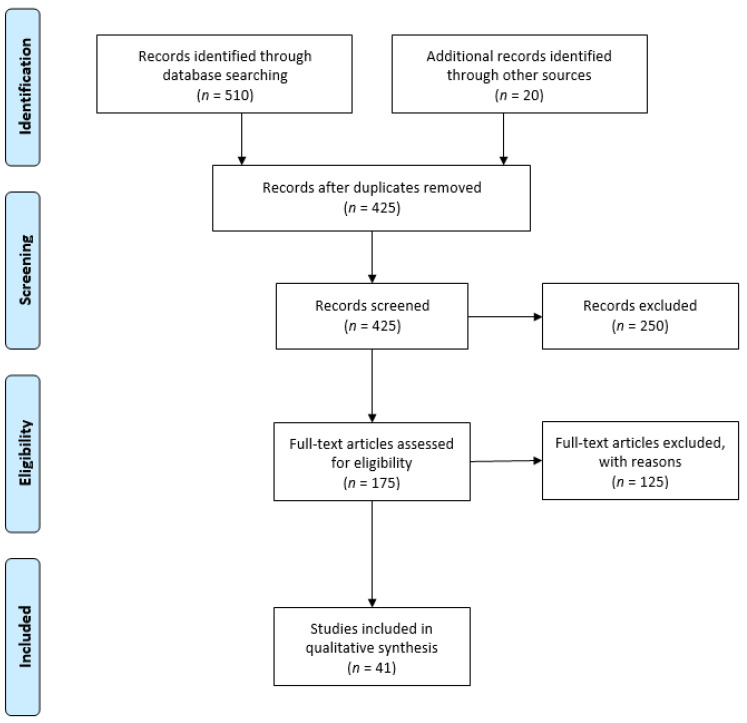
Preferred reporting items for systematic reviews and meta-analyses (PRISMA 2009) flowchart [14].

**Figure 2 jcm-10-04880-f002:**
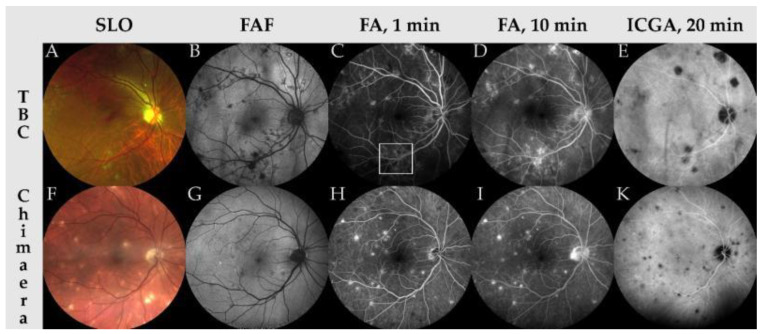
Multimodal imaging findings of patients suffering from multifocal choroiditis due to *M. tuberculosis* (TBC, **A**–**E**) and *M. chimaera* (**F**–**K**). In both scanning laser ophthalmoscopy images (SLO, **A**,**F**) choroidal granulomas can be observed. Note that panel A was acquired using an OPTOS device (OPTOS Inc., Marlborough, MA, USA) using a green and red laser, while panel F was acquired using a ZEISS Clarus device (Carl Zeiss Meditec, Inc., Dublin, CA, USA) with red, green and blue lasers, hence the difference in false color scheme. Fundus autofluorescence imaging (FAF, **B**,**G**) shows hypoautofluorescence at the location of choroidal granulomas, indicating loss of retinal pigment epithelium. In fluorescein angiography imaging at 1 min and 10 min (FA, **C**,**D**,**H**,**I**) early hypofluorescence in active lesions (panel C, inside white frame) and late-stage staining of choroidal granulomas can be appreciated. Note that in *M. chimaera*-associated multifocal choroiditis (MFC), hyperfluorescence seems to be more focused on the choroidal lesions while in *M. tuberculosis*-associated MFC the hyperfluorescences are more diffuse. In indocyanine green angiography imaging (ICGA, **E**,**K**), hypocyanescent lesions indicate the location of choroidal granulomas.

**Figure 3 jcm-10-04880-f003:**
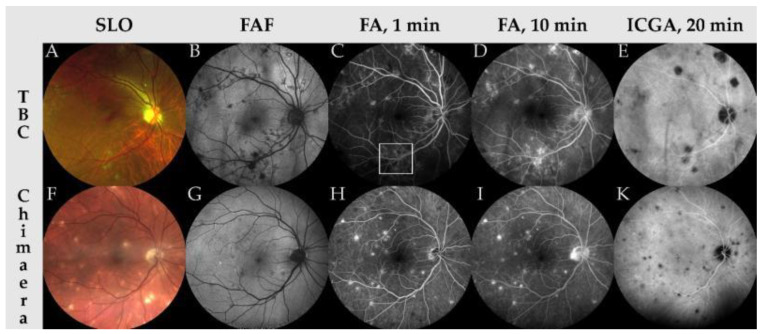
Typical imaging findings of ocular tuberculosis. Late stage fluoresceine angiography (FA; **A**) and optical coherence tomography (OCT; **B1**–near infrared fundus image; **B2**–OCT- B scan) of a 60-year-old female patient’s left eye illustrating a hot disc and macular leakage in FA which corresponds to a cystoid macular edema with sub- and intraretinal fluid in the OCT images. OPTOS false-color scanning laser ophthalmoscopy (SLO; **C1**,**C3**) and FA (**C2**,**C4**) of a 46-year-old female patient. In the SLO color image of the right eye, optic disc edema, and multiple non-continuous choroidal granulomas are visible, the vitreous seems to be inflamed in the right eye. In the FA of the right eye (**C2**), a hot disc, vascular wall hyperfluorescence and staining of choroidal granulomas can be observed. However, there are no signs of vascular occlusion. The left eye shows no sign of infection (**C3**,**C4**).

**Table 1 jcm-10-04880-t001:** Classification and comparison of choroidal lesions due to *M. tuberculosis* and *M. chimaera* based on multimodal imaging.

	*M. chimaera*		*M. tuberculosis*	
	Active Lesion	Inactive Lesion/Lesion in Regression	Active Lesion	Inactive Lesion/Lesion in Regression
Fundus Photography				
shape	ovoid to round	ovoid to round	ovoid to round	ovoid to round
border	Indistinct	well-demarcated	indistinct	well-demarcated
size	<1 optic disc diameter	<1 optic disc diameter	variable	variable
color	yellowish-white	whitish	grayish-white or yellowish	variable pigmentation
distribution	Uniform	uniform	SLC-like single or multifocal lesion	SLC-like single or multifocal lesion
**FA**				
early	Hypofluorescent	hyperfluorescent	hypofluorescent	hyperfluorescent
late	Hyperfluorescent	hyperfluorescent	hyperfluorescent	hyperfluorescent
**ICGA**	Hypocyanescent	hypocyanescent (in atrophic areas)/isocyanescent	hypocyanescent	hypocyanescent
**FAF**	hypoautofluorescent	hypoautofluorescent (in atrophic areas)/isoautofluorescent	hypoautofluorescent	hypoautofluorescent
**EDI-OCT**				
shape	full-thickness, round, well-defined borders	poorly defined margins	full-thickness, round, well defined borders	outer retinal, RPE irregularity
internal reflectivity	Hyporeflective	similar to the choroid	hyporeflective	similar to the choroid (complete resolution possible)
choroidal hypertransmission	Present	present	present	present
**OCTA**				
	CC flow reduction in the corresponding area of the lesions	CC flow reduction (in atrophic lesion)	CC flow reduction with few preserved islands in the center of the lesion	CC atrophy with visualization of medium-to-large choroidal vessels in the corresponding area

Abbreviations: M.: Mycobacterium; SLC: serpiginous like chorioiditis; FA: Fluoresceine angiography; ICGA: Indocyanine green angiography; FAF: Fundus autofluorescence; EDI-OCT: enhanced depth imaging optical coherence tomography angiography; RPE: retinal pigment epithelium; OCTA: optical coherence tomography angiography; CC: choriocapillaris.

## Data Availability

Data are available on reasonable request to the corresponding author.

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
