# Peer review of "Differences between Mycobacterium chimaera and tuberculosis Using Ocular Multimodal Imaging: A Systematic Review"

_jcm, 2021, doi:10.3390/jcm10214880_

Round 1

Reviewer 1 Report

The paper is very interesting and should contribute to this field significantly. However, I think the paper needs considerable revision - 

  1. Abstract should be rewritten and concise with a clear statement on the paper's focus
  2. consider adding the contribution to the field - in INTRODUCTION
  3. need to quantify the diagnostic results for each imaging sensors
  4. the methods should be compared in a table on each of the datasets - consider implementing the methods if necessary ..

Author Response

Dear Reviewer 1,

                    We are grateful to you for your time and constructive comments on our manuscript.

We have amended the manuscript according to your comments and valuable suggestions. Changes in the last version of the manuscript are reported as red and blue tracked changes.

Below, we also provide a point-by-point response explaining how we have addressed each of your comments.

POINT-BY-POINT RESPONSE

Comments and Suggestions for Authors

The paper is very interesting and should contribute to this field significantly. However, I think the paper needs considerable revision - 

  1. Abstract should be rewritten and concise with a clear statement on the paper's focus
  2. consider adding the contribution to the field - in INTRODUCTION
  3. need to quantify the diagnostic results for each imaging sensors
  4. the methods should be compared in a table on each of the datasets - consider implementing the methods if necessary ..

Authors’ response:

We are very grateful for the valuable comments and the possibility to implement the entire manuscript.

Regarding point 1 and 2, we have rewritten the abstract and the introduction, highlighting the contribution to the field. Indeed, to date, limited data exists about M. chimaera infections, a rare and often-lethal dis-ease. Ophthalmologists should be aware of this recently described entity and its systemic and ocular findings. They may play a key role for differential diagnosis, for monitoring treatment response and detection of recurrences after discontinuation or adjustment of treatment due to adverse drug effect.

Differential diagnosis of ocular lesions can be often challenging, thus in this review we summarized and compared the available literature on multimodal imaging data of M. chimaera infection and tuberculosis, emphasizing similarities and differences in imaging patterns between these two entities and highlighting the relevance of multimodal imaging in the management of those infections.

Regarding point 3 and 4, we have implemented the methods section and reorganized the entire review as a systematic review, according the PRISMA guidelines. A new research of the literature available has been performed once again but the results did not show new articles on the topic. A flow-chart of the literature research (Figure 1) has been designed. The title of the revision has been edited accordingly. Unluckily, no data synthesis showed in tables was possible for the heterogeneity of available data and the design of the available studies (i.e., case reports or case series). Thus, the current systematic review reports a qualitative analysis, detailed issue-by-issue below in narrative fashion.

We hope we have now clarified your main concerns and replayed to all your queries.

Looking to hear for your favorable consideration

Best regards,

All coauthors

Reviewer 2 Report

This is a very interesting review article, very well written and very well done study. I would just like to thank the authors for their effort. I have really learned a lot by reviewing all these concepts that they provide to us. Congratulations
I would just like to suggest some minor changes:

Authors duplicate/triplicate ‘’ Tuberculosis (TB) ‘’

I would try to short a litte the introduction

Author Response

Dear Reviewer 2,

                    We are grateful to you for your time and constructive comments on our manuscript.

We have amended the manuscript according to your comments and valuable suggestions. Changes in the last version of the manuscript are reported as red tracked changes.

Below, we also provide a point-by-point response explaining how we have addressed each of your comments.

POINT-BY-POINT RESPONSE

Comments and Suggestions for Authors

This is a very interesting review article, very well written and very well done study. I would just like to thank the authors for their effort. I have really learned a lot by reviewing all concepts that they provide to us. Congratulations.

I would just like to suggest some minor changes:

Authors duplicate/triplicate ‘’ Tuberculosis (TB) ‘’

I would try to short a little the introduction

Authors’ response:

We are very grateful for the positive feedback we have received. To date, limited data exists about M. chimaera infections, a rare and often-lethal disease. Ophthalmologists should be aware of this recently described entity and its systemic and ocular findings. They may play a key role for differential diagnosis, for monitoring treatment response and detection of recurrences after discontinuation or adjustment of treatment due to adverse drug effect.

Differential diagnosis of ocular lesions can be often challenging, thus in this review we summarized and compared the available literature on multimodal imaging data of M. chimaera infection and tuberculosis, emphasizing similarities and differences in imaging patterns between these two entities and highlighting the relevance of multimodal imaging in the management of those infections.

As suggested, we have edited the Introduction shorting some paragraphs. Additionally, we have implemented the final paragraph focusing on the main aim of the study.

As suggested we have used consistently the abbreviation TB for tuberculosis.

We hope to receive your favorable consideration for our paper

Best regards,

All coauthors
